# Brain Interaction during Cooperation: Evaluating Local Properties of Multiple-Brain Network

**DOI:** 10.3390/brainsci7070090

**Published:** 2017-07-21

**Authors:** Nicolina Sciaraffa, Gianluca Borghini, Pietro Aricò, Gianluca Di Flumeri, Alfredo Colosimo, Anastasios Bezerianos, Nitish V. Thakor, Fabio Babiloni

**Affiliations:** 1Department Anatomical, Histological, Forensic & Orthopedic Sciences, Sapienza University of Rome, 00185 Rome, Italy; nicolina.sciaraffa@uniroma1.it (N.S.); gianluca.diflumeri@uniroma1.it (G.D.F.); alfredo.colosimo@uniroma1.it (A.C.); 2BrainSigns, 00185 Rome, Italy; g.borghini@hsantalucia.it (G.B.); p.arico@hsantalucia.it (P.A.); 3Department Molecular Medicine, Sapienza University of Rome, 00185 Rome, Italy; 4IRCCS Fondazione Santa Lucia, 00142 Rome, Italy; 5Singapore Institute for Neurotechnology, Centre for Life Sciences, National University of Singapore, Singapore 119077, Singapore; tassos.bezerianos@nus.edu.sg (A.B.); sinapsedirector@gmail.com; (N.V.T.)

**Keywords:** mental workload, cooperation, hyperscanning, EEG, human interaction, multiple-brain connectivity

## Abstract

Subjects’ interaction is the core of most human activities. This is the reason why a lack of coordination is often the cause of missing goals, more than individual failure. While there are different subjective and objective measures to assess the level of mental effort required by subjects while facing a situation that is getting harder, that is, mental workload, to define an objective measure based on how and if team members are interacting is not so straightforward. In this study, behavioral, subjective and synchronized electroencephalographic data were collected from couples involved in a cooperative task to describe the relationship between task difficulty and team coordination, in the sense of interaction aimed at cooperatively performing the assignment. Multiple-brain connectivity analysis provided information about the whole interacting system. The results showed that averaged local properties of a brain network were affected by task difficulty. In particular, strength changed significantly with task difficulty and clustering coefficients strongly correlated with the workload itself. In particular, a higher workload corresponded to lower clustering values over the central and parietal brain areas. Such results has been interpreted as less efficient organization of the network when the subjects’ activities, due to high workload tendencies, were less coordinated.

## 1. Introduction

Most human activities are based on the interaction between two or more subjects, therefore the success of a task is based not only on high individual performance, but also on the ability to do effective teamwork. In this regard, during the last decade one of the aims of social neuroscience has been to define objective measures, mainly based on neurophysiological signals, of successful social and cooperative interaction [1,2].

During highly demanding tasks requiring cooperation among subjects, decreases in performance could be due not only to the failure of a single subject, but also to a lack of coordination within the team. In fact, it has been assessed that team coordination, defined as team members’ interaction aimed to jointly perform a key function, has positive effects on performances and affects cognitive, physiological and perceptual aspects of the involved subjects [3].

Team coordination is usually evaluated using different subjective measures. In the aviation field, for example, subject-matter experts (SMEs) are responsible for analyzing crewmembers’ interactions by means of questionnaires during a mission preparation with the aim of predicting the success of the mission itself [4]. Dodel et al. [5] measured team performance and coordination during a room cleaning task (four subjects searching and identifying threats along predefined path into a room) by using position and velocity information for three different skill levels (Novice, Intermediate and Expert). They found that (i) team coordination decreased with performance, and that (ii) Novel and Expert were significantly different in terms of team coordination and performance.

The combination of behavioral data and brain signals (electroencephalogram (EEG)) were used to identify the presence of beta-gamma complex in coordinated activities [6] and a phy-complex neuromarker of social coordination [7]. In a task as well controlled as finger movement, Tognoli et al. [7] found brain activity reduction in the occipital EEG alpha band and in EEG sensory motor rhythms during social interaction (while two subjects observed each other moving the finger), whereas coordination correlated with particular EEG rhythms, called phy, in the right centro-parietal brain area (increased phy indicates increased coordination). However, in a more realistic context, subjects are usually faced with more complex movements performing tasks under variable difficulty [8,9]. On the other hand, task difficulty could have effects on the amount of mental resources employed by the subject to deal with the task demand, that is the definition of mental workload. Mental workload can be estimate by means of behavioral and subjective measures [10], but it has also been demonstrated that mental workload variations correlate with changes in EEG spectral power, in particular an increase in the EEG theta band (4–7 Hz) over the frontal cortex, and a decrease in the EEG alpha band (8–12 Hz) over the parietal cortex [11,12,13].

Therefore, an objective evaluation of the mental workload under which the team is working could avoid a decrease in performance, which in cognitive psychology is known as the inverted “U-shape” relation between workload and performance [14], and an objective assessment of team members interaction could avoid failures due to impairment in team coordination.

To investigate objectively the interaction by means of brain signals, we used the EEG hyperscanning technique [15], which is the acquisition of EEG signals recorded in a synchronized way from two or more people involved simultaneously in a task. The aim of this forefront approach is defining how coupling in subjects’ brain activity is related to their interaction and, thanks to the EEG system portability, social interaction can be reproduced and analyzed in natural environments as required by Hari et al. [16] to investigate real social behavior. To obtain interaction-related information it was necessary to carry out multiple brain connectivity analyses, and then to explore the variation of brain network properties by means of graph analysis. The modulation of network indexes for different levels of interaction has been demonstrated both in controlled and in more ecological settings. Astolfi et al. [17] demonstrated that during a cooperative joint action task an increased number of functional connections is associated with the will of subjects to reach the same goal, unlike the condition during which the single subject was playing with the computer or alone. Moreover, functional connectivity was found among card game players of the same team when they were interested in coordinating their behavior with each other [18]. In operational environments during the landing phase a higher number of functional links between pilots is associated with the necessity of greater cooperation [19]. Sanger et al. [20] investigated the synchronization in activation during music playing: they found strengthened frontal and central connections between subjects and higher small-word network characteristic (i.e., low path length and high clustering coefficient) of both single and multiple brain network while high demand in musical coordination was requested. Nevertheless, as assessed in [21], the application of graph indexes to multiple brain networks is still an open issue.

In this work, we analyzed data gathered from couples of participants that shared collaborative tasks in the NASA-Multi Attribute Task Battery under different difficulty levels with the aim to investigate the relationship between workload and interaction effects. In particular, we hypothesized that (i) the density of inter-brain connections, as it has been associated with different levels of cooperation, increases when there is more cooperation between subjects [17], and that (ii) the strength index, necessary to assess the importance of the electrodes in the network consisting of the connections between subjects, increases as the task become more demanding [20]. Finally, we also investigate the segregation of a multiple-brain network by means of clustering coefficient to assess if there is any relationship between information transfer in the network and the task demand.

## 2. Materials and Methods

### 2.1. Participants

The selection of the participants was done in order to ensure the homogeneity of the experimental sample in terms of age, gender and educational background. In particular, informed consent was obtained from 10 healthy male participants (25 ± 3 years old), recruited on a voluntary basis, from the National University of Singapore (NUS), Centre for Life Sciences, after an explanation of the study. The experiment was conducted following the principles outlined in the Declaration of Helsinki of 1975, as revised in 2000. The study protocol has been approved by the local Ethical Committee. All the recruited participants accepted to participate to the study and each of them has been paid SG$200 to attend the whole experimentation. Additionally, the participants have been instructed to avoid alcohol, caffeine and heavy meals right before the experiments.

### 2.2. Multi Attribute Task Battery

The NASA-Multi Attribute Task Battery (MATB [22], freely available on the website reported in [23]) is a computer-based task designed by NASA to evaluate cognitive operational capability, as it could provide different tasks that have to be attended by the subjects in parallel, and each task could also be modulated in difficulty (Figure 1). By such capabilities, it is possible to investigate different cognitive phenomena requiring the simultaneous execution of actions, especially when a high level of cooperation is required (e.g., piloting an aircraft).

The MATB consists in four subtasks (panel I in Figure 1): emergency lights task; tracking task; auditory monitoring task, and; fuel managing task.

The emergency lights task simulated the system monitoring: the subject has to monitor the gauges and the warning lights by responding to the absence of the green light, to the presence of the red light, and monitor the four moving pointer dials for deviation from the midpoint.

In the tracking task the participant has to keep the cursor inside a squared target by moving the joystick. A communication task presents pre-recorded auditory messages at specific time intervals during the simulation. The goal of this subtask is to determine which messages are relevant, among irrelevant messages, and to respond by selecting the appropriate radio and frequency on the communications task window. Finally the resource management task aim is to maintain the fuel level of the main tanks at 2500 lbs by turning on or off any of the eight pumps. Pump failures occur when they are red colored.

The participants were asked to practice and learn to execute correctly the MATB for three consecutive weeks, and then the cooperation session was scheduled in the last day. During the cooperation session the 10 participants were divided in five couples (panel II in Figure 1): one subject was dealing with tracking and emergency lights task (Pilot), the other was in charge of the fuel managing task and auditory monitoring task (Co-pilot). Synchronized EEG was recorded while the couples were executing the MATB under two conditions according to the difficulty level: Easy and Hard. In the Hard condition, the difficulty of the MATB was increased by enhancing the number of events and reducing the available time to react to them with respect to the Easy condition. For example, the Easy condition time-outs were 15 s for the emergency lights, 30 s for the radio communications, pump rates of 1000 lbs/min and 500 lbs/min for the auxiliary and main tanks, respectively, and a total of radio calls. Instead, the Hard condition was characterized by time-outs of 20 s for the communication task, 5 s for the emergency lights, pump rates of 800 lbs/min and 600 lbs/min for the auxiliary and main tanks, respectively, and a total of seven radio calls.

Before starting with the experiment, a baseline condition was recorded by asking the participants to look at the MATB interface for 2 min without reacting to the events.

### 2.3. Subjective Measures

At the end of each condition, the participants were asked to fill the NASA-TLX questionnaire [24], with the aim of collecting the subjective workload perceptions across the different cooperation conditions. The NASA-Task Load Index (NASA-TLX) is a widely-used assessment tool that rates perceived workload in order to assess a task, a system, a team’s effectiveness or other performance aspects. The total workload score, ranging from 0 to 100, was calculated as a weighted combination of the six factors of the questionnaire (Mental Demand, Physical Demand, Temporal Demand, Performance, Effort and Frustration) in order to take into account all the possible aspects interplaying in the workload level definition.

### 2.4. Performance Indexes

Four performance indexes have been defined, one for each subtask. In particular, the index for the tracking task has been defined by considering the complement of the ratio between the cursor’s distance got by the subject and the maximum of this distance (fixed) from the center of the screen, as reported in the following formula:(1)$$Tracking\;Task\;Index=\left(1-\frac{Cursor\;Position}{Max\;Distance}\right)\times 100$$

The indexes of the auditory monitoring task and emergency light task have been defined as a linear combination of accuracies in terms of correct answers (e.g., correct radio or frequency selected) and the ratio between the subject’s reaction time and the maximum time for answering; then, the results have been multiplied by 100 in order to obtain a percentage, as reported in the following formulas:(2)$$Auditory\;Task\;Index=\left(\%\;Correct\;Communication+\frac{Mean\;RT}{Available\;RT}\right)\times 100$$
(3)$$Emergency\;Light\;Task\;Index=\left(\%\;Correct\;Lights+\frac{Mean\;RT}{Available\;RT}\right)\times 100$$

Finally, the index for the fuel managing task has been defined as the mean value of the fuel level in the main tanks and then multiplied by 100, as reported in the following formula:(4)Fuel Managing Index=(% Fuel Stability Tank A+ % Fuel Stability Tank B)×100

In order to get a global Performance Index, the average of the previous indexes has been calculated.

### 2.5. EEG Signal Pre-Processing

The EEG was recorded by a digital monitoring system (ANT Waveguard system, ANT Neuro, Enschede, Netherlands) with a sampling frequency of 256 Hz. All the 30 EEG electrodes for each subject were referred to both earlobes, grounded to the AFz channel, their impedances were kept below 10 kΩ, and the recording systems was synchronized to avoid delay between the signals recorded from two different subjects.

The EEG signal was firstly band-pass filtered with a fifth-order Butterworth filter (low-pass filter cut-off frequency: 45 Hz, high-pass filter cut-off frequency: 1 Hz), and then it was segmented into epochs of 1 s. Each EEG epoch with amplitude higher than ±80 μV was removed in order to have an artifact-free EEG dataset from which compute the workload and estimate the parameters for connectivity analysis. All the analysis tools have been implemented in Mathworks MATLAB.

### 2.6. EEG—Based Workload Index

From the artifact-free EEG dataset, the power spectral density (PSD) was calculated for each EEG epoch using a Hanning window of the same length of the considered epoch (1-s length means 1 Hz of frequency resolution). Then, the EEG frequency bands were defined accordingly with the individual alpha frequency (IAF) value estimated for each subject [25]. Since the alpha peak is mainly prominent during rest conditions, the participants were asked to keep their eyes closed for a minute before starting with the experiment. Such condition was then used to estimate the IAF value specifically for each subject. In particular, the theta [(IAF − 6), (IAF − 2)] and alpha brain activities [(IAF − 2), (IAF + 2)] were considered over the EEG frontal (F7, F3, Fz, F4, and F8) and parietal (P7, P3, Pz, P4, and P8) channels in order to assess the workload. In fact, it was widely demonstrated how such EEG bands are the most correlated to mental workload variations [9,11,26,27]. In this regard, a mental workload index (MWI) is defined as:(5)$$MWI=\frac{PSD_{\theta_{F}}}{PSD_{\alpha_{P}}}$$
where the $PSD_{\theta_{F}}$ represents the PSD in the theta band estimated over the frontal brain areas and the $PSD_{\alpha_{P}}$ is the PSD in the alpha band estimated over the parietal brain areas. The mental workload index in Easy and Hard condition have then been normalized on the baseline condition.

### 2.7. Multiple-Brain Connectivity

Artifacts-free epochs acquired simultaneously from each couple underwent a multiple-brain connectivity analysis. This analysis provide the information flows exchanged between scalp areas within each single subject (intra-connections) and between the couple (inter-connections), namely, scalp areas belonging to the two subjects in pair. Multiple-brain connectivity estimation is computed through the partial directed coherence (PDC) estimator adapted to the multi-subject case [18].

PDC is a full multivariate spectral measure used to determine the directed influences among any given pairs of signals in a multivariate data set. This estimator represents a frequency version of the concept of Granger causality [28].

Let Y be a set of signals, obtained from non-invasive EEG recordings:(6)$$Y=\left[y_{1}\left(t\right),y_{2}\left(t\right),\ldots ,y_{N}\left(t\right)\right]$$
where *t* refers to time and *N* is the number of considered signals. Supposing that the following multivariate autoregressive (MVAR) process is an adequate description of the dataset Y:(7)$$\overset{p}{\underset{k=0}{\sum }}A\left(k\right)Y\left(n-k\right)=E\left(n\right),\;with\;A\left(0\right)=I$$
where Y(t) is the data vector in time, *E*(*t*) = [*e*1(*t*),…, *e*n(*t*)] is a vector of multivariate zero-mean uncorrelated white noise processes, *A*(1), *A*(2),…, *A*(*p*) are the *NxN* matrices of model coefficients and *p* is the model order, usually chosen by means of the Akaike Information Criteria (AIC) for MVAR processes [29]. Equation (7) can be transformed to a frequency domain by implementing the z-transformation of each term:(8)A(f)Y(f)=E(f)
where A(f) represents the frequency version of A(k) along the *p* lags considered in the estimation.

It is possible to define the PDC estimator as follows:(9)$$\pi_{ij}\left(f\right)=\frac{{\left|A_{ij}\left(f\right)\right|}^{2}}{\sum_{m=1}^{N}{\left|A_{mj}\left(f\right)\right|}^{2}}$$
where $A_{ij}\left(f\right)$ represents the frequency version of *ij* coefficient of multivariate autoregressive (MVAR) model used for modeling the dataset under investigation.

In the present paper, we employed a different normalization of PDC estimator called generalized PDC (gPDC) recently introduced to circumvent the numerical problem associated with time series scaling [30]. gPDC can be defined as follows:(10)$$g\pi_{ij}\left(f\right)=\frac{{\left|A_{ij}\left(f\right)\right|}^{2}\sigma_{i}^{-2}}{\sum_{m=1}^{N}\frac{1}{\sigma_{m}^{2}}{\left|A_{mj}\left(f\right)\right|}^{2}}$$
where $\sigma_{i}$ refers to the variances of the residuals E. The choice of gPDC was justified by results obtained in [31] where the insensitivity of gPDC to scale differences in the amplitude of signals to be fed into the MVAR model was highlighted.

Multiple-brain connectivity patterns were estimated for each experimental condition (Baseline, Easy, Hard) and each pair included in the study on 20 electrodes (F3, F4, F7, Fz, F8, FC5, FC6, C3, Cz, C4, CP5, CP6, T7, T8, P3, Pz, P4, O1, O2, Oz). The obtained gPDC values were averaged within three frequency bands (Theta: 4–8 Hz, Alpha: 8–12 Hz, Beta: 12–25 Hz).

In order to obtain Grand Average Statistical Connectivity Maps, we performed a paired *t*-test between connectivity patterns related to the target experimental conditions (Easy and Hard) and those referring to the corresponding baseline. The statistical comparison was repeated for each of the three frequency bands.

### 2.8. Multiple-Brain Graph Theory

Connectivity matrices estimated for each subjects pair in Easy and Hard condition were statistically compared, at single-subject level, with threshold values extracted by applying the 95th percentile on the baseline-related gPDC distribution built among the experimental group. Such comparison made it possible to obtain for each couple, each experimental condition and each EEG band, an adjacency matrix on which graph theory indices were calculated.

The density of inter-brain connections in each condition was computed as the ratio between the number of existing inter-connections, and the number of all possible connections between subjects.

Furthermore, we decided to consider indices providing local information about the level of interaction between the two subjects: strength and clustering coefficients. According to [32] strength values are informative about the brain processes and clustering coefficients about the local information transfer.

The strength is defines as [33]: (11)$$k_{i}^{w}=\underset{j\in N}{\sum }w_{ij}$$
where *N* is the set of all nodes in the network; $w_{ij}$ is the connection weight. According to gPDC normalization, values of weights are ranging between 0 and 1. The weight represent the value of the direct connection, that is, *i* vertex exerts influence on *j* vertex.

The clustering coefficient of a vertex in a network is the fraction of triangles around a node and is defined as [34]:(12)$$C_{i}=\frac{2t_{i}}{k_{i}\left(k_{i}-1\right)}$$
where $C_{i}$ is the clustering coefficient of node *i* that belongs to set *N*, $k_{i}$ represents total degree and $t_{i}$ represents the number of triangles around the node *i*.

In this case we are interested only in evaluating the connections between subjects, therefore the strength and the clustering coefficient were computed on the between brain networks. Then the two indices were averaged across four main scalp areas: frontal (F3, Fz, F4); central (C3, Cz, C4); parietal (P3, Pz, P4), and; occipital (O1, O2, O3).

These indices were then statistically compared between Easy and Hard conditions in three frequency bands (Theta, Alpha and Beta) for the whole population, as no significant differences in terms of workload perception were found between the Pilot and Co-Pilot groups. Moreover, their values were correlated (Pearson’s correlation) with those of the mental workload index in the considered experimental group.

## 3. Results

### 3.1. Performance and Mental Workload Index

Moving from the Easy to Hard condition, a significant decrease in the global performance index (*p* = 0.014) was found (panel a in Figure 2). The comparison between workload perceived by the Pilot and Co-Pilot groups on the same condition was not significantly different both in Easy (*p* = 0.625) and in Hard (*p* = 0.438) according to the Wilcoxon test. On the contrary, considering the entire experimental population, the perceived workload in the Hard condition was significantly higher (*p* = 0.000018) than the Easy (panel b in Figure 2). The same trend (panel c in Figure 2) was showed by the computed mental workload index (*p* = 0.059).

### 3.2. Multiple-Brain Connectivity

#### 3.2.1. Grand Average

Figure 3 represents the Grand Average Connectivity maps obtained from the statistical contrast by means of paired *t*-test between Easy and Hard conditions, respectively, against the baseline condition. In each couple, the scalp model on the left represents subject doing audio and fuel task (Co-Pilot), whilst the scalp model on the right represents subjects doing tracking and lights task (Pilot). Only significant (*p* < 0.05) connections were reported by means of arrows whose color codifies the averaged value of the connection. The value of inter-brain density (D) for each condition is also reported. It has to be noticed the lower number of inter-connections in the Hard condition with respect to the Easy one.

#### 3.2.2. Inter-Brain Density

The inter-brain density showed an increasing trend in the Hard condition in the EEG theta and alpha band, while a decrement in EEG beta band, but none of these differences were statistically significant (Table 1).

#### 3.2.3. Strength

Figure 4 shows the strength averaged in each brain area of interest for the two experimental conditions. It can be seen that there are higher values in the Hard condition than in the Easy one. Such a trend was significant over the frontal brain area in the theta band, and over all the considered brain areas in the alpha band (Table 2).

#### 3.2.4. Clustering Coefficient

The clustering coefficient averaged in the brain areas of interest has been correlated with the mental workload index. Significant negative correlations were found for the clustering coefficient averaged over the central brain area in the Hard condition, both in the alpha (*R* = −0.772, *p* = 0.008) and in the beta band (*R* = −0.784, *p* = 0.007). Moreover, the clustering coefficient averaged over the parietal brain area (Figure 5) correlated negatively with the workload index in the beta band (*R* = −0.690, *p* = 0.026). No significant correlation was found in the Easy condition.

## 4. Discussion

In this study, the cooperation between humans performing operational tasks under different difficulty levels has been investigated by means of subjective, behavioral and physiological measures. The hyperscanning approach that was used made it possible to apply the multi-subject connectivity analysis that, as demonstrated in previous work [17,19], made it possible to discriminate the degree of cooperation on the basis of the corresponding brain network properties.

Firstly, we analyzed the effects of different task difficulty on the subjects: a more demanding condition caused a significant reduction in performance with respect to the Easy condition. Subjective measures (the NASA-TLX questionnaire) reflected significant increments in the Hard condition, showing that it was actually perceived as more demanding. The neurometric (i.e., mental workload index) showed a marginally significant (*p* = 0.059) increase in the Hard condition, coherently with the cognitive psychology literature that demonstrated an inverted relationship between mental workload and performance after exceeding an optimal threshold [11].

Secondly, the multiple-brain connectivity analysis was carried out with the aim to provide the causal influences between activated brain areas, not necessarily belonging to the same subject. Each of this area can be describe as a node and the causal relationship by an edge, so the connectivity matrix can efficiently describe a network. Qualitatively, the number of connections between subjects (inter-connections), significantly different (*p* < 0.05) from baseline condition, were higher in Easy than in Hard (Table 1) condition, but there was not a significant difference in inter-brain density value between the two conditions (Table 1). In previous works differences in the number of inter-brain connections had been related to different levels of interaction. However, in this work no differences have been found, probably because different levels of workload of the same task did not lead to different levels of coupling activities between participants’ brain. Nevertheless, there was a modulation from Easy to Hard condition of local features describing the network in subsets of electrodes representative of the frontal, central, parietal and occipital brain areas.

Higher strength due to inter-connections during a more demanding task (Hard) in the EEG alpha band (and also in theta band over the frontal brain area) can be explained as a higher involvement of those areas in both participants, as the strength is explanatory of brain processes [32]. Then, while there was not a modulation in terms of number of connections involving the two subjects, in the case of a more difficult task the average value of the inter-connections was significantly higher than in an Easy one, as expected.

Finally, during the Hard condition the subjects who exhibited lower workload also had lower clustering coefficient in the central and parietal brain areas, as reported in the negative correlation (Figure 5). Since high clustering is associated with robustness of a network, it can be assumed that higher workload caused impairment in local connectivity in the beta band for both parietal and central areas, and in the alpha band for the central area. Diminished local clustered connectivity in those areas indicates less efficiency in information transfer and a behavior closer to a random network [35]. From the mental workload perspective, the significant correlation in case of the Hard condition, but not in case of the Easy one, could be assumed to be a sign of network properties alteration as the subjects were closer to the overload point.

These results in terms of bands and areas of interest are coherent with previous knowledge both about the role of EEG alpha and beta bands [6], and about the central (sensory-motor system), and parietal (mirror neuron system) brain areas in human interaction and coordination [7].

## 5. Conclusions

The proposed study demonstrated that different workload demands impacted differently on the individual team member both in terms of performance, and experienced workload, but also on the whole team in terms of different local network’s features, describing the brain activities that were Granger-caused to each other by the two subjects.

Strength value was proven to be significantly higher for a more difficult task in the alpha band, while clustering coefficient showed negative correlation with workload index in the Hard condition in bands and areas of interest, respectively, alpha and beta EEG bands and central and parietal brain areas, as expected in coordination and interaction phenomena.

To sum up, two subjects sharing a task can be seen as a system where mental resources required for each subject and some features representing the whole system are not independent: in the context of social neuroscience this relationship could be a fertile ground for the definition of a neuromarker of efficient interaction.

In this study only a limited number of indexes was used with particular focus on local properties. Nevertheless, in the future studies, a more complete description of global network properties and different kind of motifs, aside from clustering, will be computed to better investigate the relationship between task difficulty and interaction. An online analysis of EEG signals will be used to follow how both neuromarkers of interaction and workload vary over time in order to predict performance decreases caused by lack of interaction and/or overload condition.

As the EEG hyperscanning approach makes it possible to examine real social behavior in a real environment, by using a higher number of electrodes it could be also possible to reconstruct, using functional imaging methods based on EEG, the cortical and sub-cortical brain activations with high temporal resolution to investigate how interaction processes and behaviors are implemented by brain system.

Finally, although the size of the experimental group was low, this work showed convergent findings with previous research and the possibility to correlate information of multiple-brain system with workload assessment to have a wider view of how coordinated interaction is affected by different workload demands and, in turn, how it can affect performance. Moreover, since the considered experimental task (NASA MATB) is freely available, further experiments will be run or replicated in order to enlarge the experimental group and validate the results described in the presented work.

## Figures and Tables

**Figure 1 brainsci-07-00090-f001:**
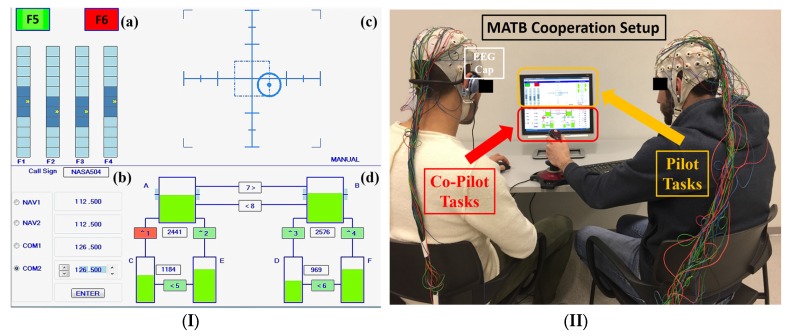
NASA-Multi Attribute Task Battery (MATB). (**I**) Interface: (**a**) Emergency lights task; (**b**) Tracking task; (**c**) Auditory monitoring task, and; (**d**) Fuel managing task. (**II**) Participants shared the MATB task. In particular, the Pilot (on the right) performed the tracking and emergency lights tasks, while the Co-Pilot (on the left) dealt with the fuel managing and the auditory monitoring tasks.

**Figure 2 brainsci-07-00090-f002:**
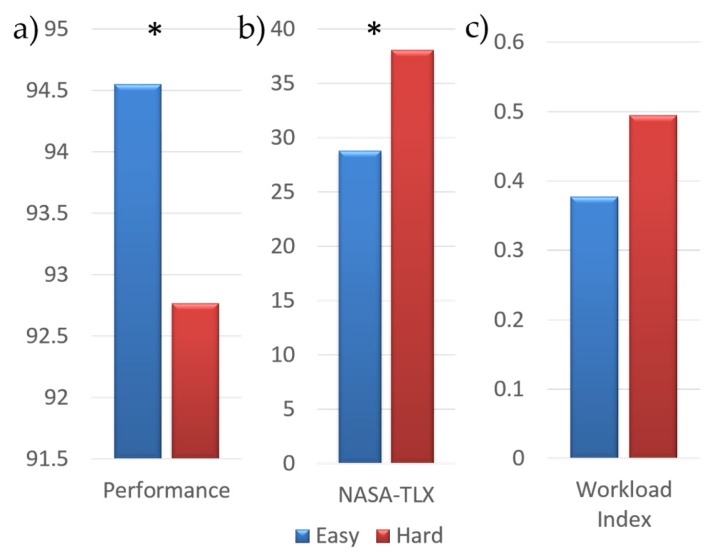
Bar diagram reporting the performance (**a**), NASA-Task Load Index (NASA-TLX) (**b**) and workload index (**c**) value obtained in Easy (blue bar) and Hard (red bar) conditions. The symbol * points out a significant difference between the considered conditions (*p* < 0.05).

**Figure 3 brainsci-07-00090-f003:**
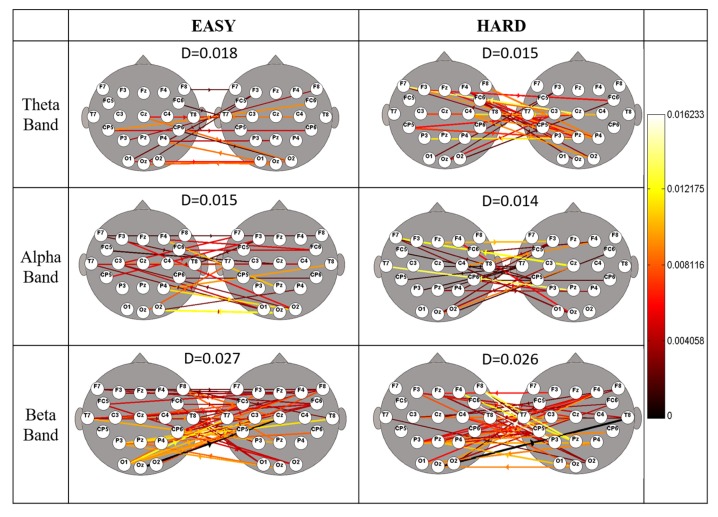
Grand Average Connectivity Maps obtained comparing Easy (first column) and Hard (second column) conditions with baseline condition in Theta (first row), Alpha (second row) and Beta (third row) conditions. The patterns are reported on a 2-D scalp model seen from above with the nose pointing to the upper part of the page. Only significant connections were reported (paired *t*-test, *p* < 0.05). The colour and diameter of the arrows code for the averaged strength were obtained within the experimental group. The value D is the inter-brain density.

**Figure 4 brainsci-07-00090-f004:**
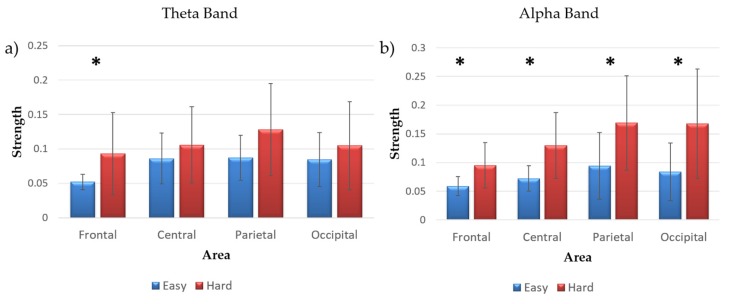
Bar diagram reporting the strength value obtained in Easy and Hard conditions over frontal, central, parietal and occipital brain areas, in theta (**a**) and alpha (**b**) electroencephalogram (EEG) bands. The symbol * points out statistical difference between the considered conditions (paired *t*-test, *p* < 0.05).

**Figure 5 brainsci-07-00090-f005:**
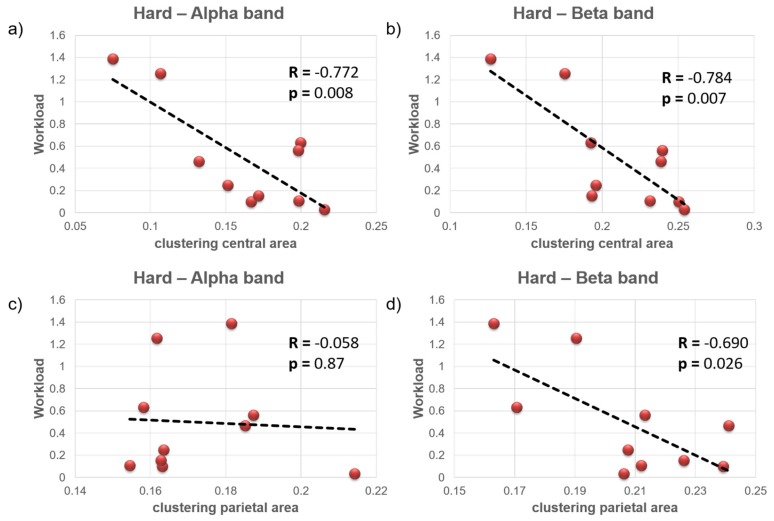
Scatterplot reporting the trend of workload index over clustering index for central brain area in the alpha (*R* = −0.772, *p* = 0.008) and beta (*R* = −0.784, *p* = 0.007) EEG bands (**a**,**b**), and for parietal brain area in the alpha (*R* = −0.058, *p* = 0.87) and beta (*R* = −0.690, *p* = 0.026) EEG bands (**c**,**d**). Plots refer to the Hard condition.

**Table 1 brainsci-07-00090-t001:** Inter-brain density.

	Easy	Hard	*p*-Value
Theta	0.074 ± 0.005	0.094 ± 0.024	0.104
Alpha	0.080 ± 0.019	0.105 ± 0.020	0.077
Beta	0.124 ± 0.029	0.117 ± 0.019	0.703

**Table 2 brainsci-07-00090-t002:** *t*-Test *p*-value Easy versus Hard condition.

Area	Theta	Alpha
Frontal	0.043 *	0.039 *
Central	0.447	0.014 *
Parietal	0.070	0.010 *
Occipital	0.305	0.009 *

The symbol * points out statistical difference between the considered conditions (paired *t*-test, *p* < 0.05).

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
