# Peer review of "Brain Interaction during Cooperation: Evaluating Local Properties of Multiple-Brain Network"

_brainsci, 2017, doi:10.3390/brainsci7070090_

Round 1

Reviewer 1 Report

In this paper an interesting approach of investigating performance and mental effort in easy and hard conditions of a cooperative task is presented. As the focus of the study lies on the complete team (2 persons), multiple-brain connectivity analysis was conducted. It has been found that strength of the connections well as clustering coefficients correlated with workload. 

For better understandability, clear hypotheses should be formulated and referred to throughout the article. 

The section about the research design should be extended as for example the performance indexes are unclear and the MATB could be described in more detail in order to better understand the set-up and indexes. Furthermore, it should be outlined why only male subjects have been chosen and how the roles of pilot and copilot have been assigned.

5 pairs of data is highly critical for conducting A paired t-test. If possible, more data should be collected. Furthermore, the report of statistical values should be consistent throughout the paper and a significance level of 5% should be handled at all times. 

In the discussion section the broader implications of the present findings should be outlined and they should be more directly linked to previous research findings. 

Grammar and spelling needs major revisions.  

Author Response

We are grateful to the reviewer for the valuable comments, which indeed helped us to improve the quality of the manuscript. We provided a point-to-point reply to the reviewer’s requests.

For better understandability, clear hypotheses should be formulated and referred to throughout the article

-       We apology with the author for the lack of clarity. As suggested, at the end of the introduction section we improved the description of experimental hypothesis to enhance the understandability of the whole work. 

The section about the research design should be extended as for example the performance indexes are unclear and the MATB could be described in more detail in order to better understand the set-up and indexes. Furthermore, it should be outlined why only male subjects have been chosen and how the roles of pilot and copilot have been assigned.

-       We would like to thank the Reviewer for the comment. As suggested, in the revised version we added more detailed descriptions of the MATB subtasks in order to improve the understandability of the performance indices. In addition, we decided to involve in the study just male subjects for recruiting facilities, and avoiding additional factors to be considered in the statistical analysis (i.e. gender). Concerning the role of the participants, the pilot/copilot have been randomly assigned.

5 pairs of data is highly critical for conducting a paired t-test. If possible, more data should be collected. Furthermore, the report of statistical values should be consistent throughout the paper and a significance level of 5% should be handled at all times. 

-       We apology with the reviewer for the lack of clarity. Actually, paired t-test has been conducted between 2 groups of 10 subjects as we assessed that there were no differences between pilot and co-pilot users in terms of perceived workload. In the revised version, we provided a non parametric test (Wilcoxon rank test) to assess this latter difference. In addition, as suggested, the significant level information has been included all times.

In the discussion section the broader implications of the present findings should be outlined and they should be more directly linked to previous research findings. 

-       As suggested by the reviewer, in the discussion section we added a more complete description about broader implications of the reported results. In addition, the presented work has been better contextualized with respect previous related works. 

Grammar and spelling needs major revisions.  

-       As suggested by the reviewer, the work has deeply revised from a grammar and spelling typos.

Reviewer 2 Report

I have to admit that I really like this paper. It covers a very innovative field of study and very promising path in cognition and human machine interactions. However I must comment on several minor issues I have detected.

Information about participants is incomplete. Demographic data should be provided as well as other aspects relevant to the sample (education, age of each participant, mean and SD, gender, etc.) We have detected in our studies that previous experience with video games for example may influence on the perceived workload. The sample information issue is very important in this manuscript and should be revise before its publication. Perhaps a table with all this data could be helpful and enough. Using the word "subject" should be avoided according to recent guidelines. 

It is not well defined the difference between hard and easy conditions. Authors should specify what easy and hard means in the tasks they proposed. 

It is not well specified what type of software was used to analyze the EEG recordings. The ANT software? EEGLAB?. This could be good to know. EEGLAB/MATLAB is standard in these type of studies.

It is not clear when the authors refer to workload if they differentiate all the time between mental and physical workload. NASA TLX offers the opportunity to differentiate these components as well as frustration, etc. We have used NASA TLX before and it can be a very helpful tool to differentiate these components in human factors situations. 

What about the NASA tests licenses? no need for that? what about the references for these tests? This could be a problem. 

In summary a good paper with some weak points that should be addressed. 

Author Response

We are grateful to the reviewer for the valuable comments, which indeed helped us to improve the quality of the manuscript. We provided a point-to-point reply to the reviewer’s requests.

Information about participants is incomplete. Demographic data should be provided as well as other aspects relevant to the sample (education, age of each participant, mean and SD, gender, etc.) We have detected in our studies that previous experience with video games for example may influence on the perceived workload. The sample information issue is very important in this manuscript and should be revise before its publication. Perhaps a table with all this data could be helpful and enough. Using the word "subject" should be avoided according to recent guidelines.

-           We would like to thank the Reviewer for the comment. We recruited 10 healthy male participants (25±3 years old)), on a voluntary basis, from the National University of Singapore (NUS). In particular, they were students at Singapore Institute for Neurotechnology (SiNAPSE), therefore they had homogeneous backgrounds. In addition, an inclusion criterion to participate to the experimental study was a high level of experience with video games. As suggested by the reviewer, we added such information within the revised manuscript in the “Participant” section.

It is not well defined the difference between hard and easy conditions. Authors should specify what easy and hard means in the tasks they proposed.

-           We apology for the lack of clarity. As suggested by the reviewer, in the revised version of the manuscript, we added further information about the distinction between the Hard and Easy difficulty levels, and specified that they referred to the proposed task.

It is not well specified what type of software was used to analyze the EEG recordings. The ANT software? EEGLAB?. This could be good to know. EEGLAB/MATLAB is standard in these type of studies.

-           We apology with the reviewer for the lack of clarity. The EEG was recorded by a digital monitoring system (ANT Waveguard system) and then analyzed by means of tools developed by the Authors in MATLAB code.

It is not clear when the authors refer to workload if they differentiate all the time between mental and physical workload. NASA TLX offers the opportunity to differentiate these components as well as frustration, etc. We have used NASA TLX before and it can be a very helpful tool to differentiate these components in human factors situations.

-           We thanks the reviewer for the suggestion, in this study we did not differentiate between mental and physical demand, and the total workload score was calculated as a weighted combination of the six factors of the questionnaire (Mental Demand, Physical Demand, Temporal Demand, Performance, Effort and Frustration). In fact, we wanted to characterize the workload as the interaction of different factors as it is usually defined, therefore taking into account all the possible aspects when dealing with a task.

What about the NASA tests licenses? no need for that? what about the references for these tests? This could be a problem.

-           As suggested, we provided more information and references in the revised version. License for the NASA-TLX is not necessary because it is freely available on the official website (https://humansystems.arc.nasa.gov/groups/tlx/index.php).